# Highly Proton-Conducting Membranes Based on Poly(arylene ether)s with Densely Sulfonated and Partially Fluorinated Multiphenyl for Fuel Cell Applications

**DOI:** 10.3390/membranes11080626

**Published:** 2021-08-15

**Authors:** Tzu-Sheng Huang, Tung-Li Hsieh, Chih-Ching Lai, Hsin-Yi Wen, Wen-Yao Huang, Mei-Ying Chang

**Affiliations:** 1Department of Photonics, National Sun Yat-Sen University, Kaohsiung 80424, Taiwan; zxp86133@gmail.com (T.-S.H.); jackko61010@gmail.com (C.-C.L.); 2General Education Center, Wenzao Ursuline University of Languages, Kaohsiung 80793, Taiwan; tunglihsieh@gmail.com; 3Department of Green Energy and Environmental Resources, Chang Jung Christian University, Tainan City 71101, Taiwan; hywen@mail.cjcu.edu.tw

**Keywords:** poly(arylene ether)s, ionomers, proton exchange membranes, fuel cell

## Abstract

Series of partially fluorinated sulfonated poly(arylene ether)s were synthesized through nucleophilic substitution polycondensation from three types of diols and superhydrophobic tetra-trifluoromethyl-substituted difluoro monomers with postsulfonation to obtain densely sulfonated ionomers. The membranes had similar ion exchange capacities of 2.92 ± 0.20 mmol g^−1^ and favorable mechanical properties (Young’s moduli of 1.60–1.83 GPa). The membranes exhibited considerable dimensional stability (43.1–122.3% change in area and 42.1–61.5% change in thickness at 80 °C) and oxidative stability (~55.5%). The proton conductivity of the membranes, higher (174.3–301.8 mS cm^−1^) than that of Nafion 211 (123.8 mS cm^−1^), was the percent conducting volume corresponding to the water uptake. The membranes were observed to comprise isolated to tailed ionic clusters of size 15–45 nm and 3–8 nm, respectively, in transmission electron microscopy images. A fuel cell containing one such material exhibited high single-cell performance—a maximum power density of 1.32 W cm^2^ and current density of >1600 mA cm^−2^ at 0.6 V. The results indicate that the material is a candidate for proton exchange membranes in fuel cell applications.

## 1. Introduction

The fuel cell is a type of energy conversion device, and the prototype fuel cell, which converted chemical energy into electrical energy through an electrochemical mechanism, was produced at the beginning of the 19th century [1,2,3]. Due to the world’s thirst for energy, fuels such as coal, oil, and gas are constantly consumed, and the generation of greenhouse gases has become an environmental concern. Hydrogen energy—a clean energy resource, producing little pollution (no CO_2_), small emissions, and low noise—is being developed by countless researchers in this era of the pursuit of green energy [4]. On the other hand, hydrogen is produced following the opposite principle of fuel cells, according to which it is produced from water splitting and used as a carrier for energy storage. Additionally, the water electrolysis unit can be used in combination with a fuel cell unit to store intermittent or overflow energy (ex. solar, wind, waste heat, or nuclear). In addition, fuel cells can be used for energy storage applications, which is how they differ from, or have advantages over, lithium-ion batteries [5] and redox flow batteries [6]. In addition to the basic ion selectivity differences, membrane performance needs to be able to perform well at extreme temperatures (i.e., −40 to 60 °C) in order to ensure environmental sustainability. Correspondingly, they may be affected during cold/hot shock cycling and charge/discharge cycling to maintain good ionic conductivity, mechanical properties, electrochemical and thermal stability [5,6]. However, their future development will depend on functional molecular designs while meeting the key economic and technical drivers of cost, power density, efficiency, and durability.

Among them, proton-exchange membrane (PEM) fuel cell using functional polymer materials as the core has been widely developed. In addition to having high proton conductivity and favorable mechanical properties, PEMs meet various performance requirements—low sensitivity to humidity, low oxidant permeability, and adequate electrochemical, chemical, thermal, and dimensional stability—If a satisfactorily low-cost and functional polymer material is to be manufactured [7,8,9,10]. PEMs have four major types—perfluorinated sulfonic acid (PFSA), fluorinated, hydrocarbon (HC)-based, and functional PEMs, all of which are widely researched. PFSA polymer materials (such as the Nafion, Flamion, Aciplex, and Dow series membranes) have excellent performance and are widely used in commercial applications [11]. However, the high pollution and external costs due to perfluorinated compounds have prompted many research teams to investigate alternatives. By contrast, HC-based PEMs are the most environmentally friendly because they are completely free of halogens, but they rarely combine high proton conductivity and favorable mechanical or other crucial properties. Functional sulfonated poly(arylene ether)s have higher thermostability and increased proton conductivity at temperatures above 80 °C and in low humidity. Poly(arylene ether)s have sulfonated derivatives of various forms, such as poly(arylene ether ketone)s [12,13,14,15], poly(arylene sulfide)s [16,17], polyimides [18,19,20], and polybenzimidazoles [21,22,23], which have been extensively investigated because of their high thermal stability, favorable chemical stability, appropriate mechanical properties, and low production cost, as well as the easy adjustment of their molecular structure [24,25]. Countless efforts have been made to synthesize new polymer architectures with conductive components (moieties); nevertheless, the influence on the morphology at many scales is not clear; therefore, PEMs have high proton conductivity, proper hygroscopicity, and sufficient mechanical properties, but at the same time, are still under study [9].

In general, sulfonated HC-based polymers have a wide diversity of morphology types because hydrophobic and hydrophilic moieties are multiscale distributed by structural effects. These are microphase separations to form proton transport channels. As reported widely, the chemistry and architecture of PEMs have effects on proton conductivity, they can adjust the structural design of polymer composites to improve by the ion exchange capacity and microphase separation [26]. We recently reported a series of poly(arylene ether)s with a high-free-volume multiarylbenzene (MAB) structure in the polymer backbone and a trifluoromethylphenyl side chain; this series has high proton conductivity, mechanical and dimensional stability, and film-forming ability [27,28,29]. Adamski et al. reported two classes of HC-based PEMs, sulfonated phenylated poly(phenylene) biphenyl and sulfonated poly(arylene ether), with branching multiphenylated structures. They described the relationship between densely sulfonated polymers and water uptake and transport, aiding in the design of next-generation solid polymer electrolytes [30]. Lee et al. synthesized a series of ethynyl-terminated sulfonated–fluorinated poly(arylene ether) random copolymers, including benzenesulfonate, hexafluoroisopropylidene, and perfluorobenzene derivatives; molar volume per charge (MCV), percent conducting volume (PCV), and derivative parameters were introduced for evaluation of the membrane properties and comparison of cross-linked poly(arylene ether)s with Nafion membranes [31]. The hydrophilic and hydrophobic substituents of the hydrophobic polymer backbone strongly influenced the microphase separation. With dense sulfonation, a superhydrophilic domain can be produced, and local halogenation or perfluoroalkane substitution can create a superhydrophobic domain in the structure. Furthermore, the trifluoromethyl-substituted group is a strongly electron-withdrawing group and can deactivate the nucleophilic aromatic substitution (S_N_Ar) reaction of the sulfonating agent, creating favorable localization for hydrophobicity [13,32,33]. However, compatibility challenges may occur when the polarities of the components are extremely different, regardless of whether the monomers are polymerized through physical blending or preacidification.

Herein, we report a series of novel poly(arylene ether)s synthesized using a multiphenylated difluoro monomer with a trifluoromethylphenyl side chain and three types of multiphenylated bisphenol monomer. Sulfonated poly(arylene ether)s were prepared through treatment with chlorosulfuric acid, with the trifluoromethyl (-CF_3_) substituent expected to inhibit the S_N_Ar reaction in specific moieties, effectively dividing the local, densely sulfonated moieties. After sulfonation, bipolar domains form in the molecules, which become partially fluorinated sulfonated poly(arylene ether)s with highly efficient proton transport. Furthermore, the free volume effect and hydrogen bond cohesion of bisphenol monomers with different degrees of sulfonation provide corresponding water sorption ability and preserve high mechanical stability. The synthesis of sulfonated poly(arylene ether)s was confirmed using proton nuclear magnetic resonance (^1^H NMR) and Fourier transform infrared (FTIR) spectroscopy. PEMs were prepared through solution casting with dimethyl sulfoxide (DMSO), and their properties—water uptake, dimensional stability, mechanical strength, proton conductivity, morphology, and single-cell performance—were determined.

## 2. Materials and Methods

### 2.1. General Methods

2,5-Bis(4-bromophenyl)-3,4-bis(3-(trifluoromethyl)phenyl)cyclopenta-2,4-dienone (DTF-EO), 2″,3″,5″,6″-Tetraphenyl-[1,1′:4′,1″:4″,1′′′:4′′′,1′′′′-quinquephenyl]-4,4′′′′-diol (9B-DO), 4-Bromo-4′-(4-bromophenyl-3′,5′-diphenyl-1,1′;2′1″-terphenyl (6B-DB), and 3′,6′-bis(4-bromophenyl)-1,1′:2′,1″-terphenyl (5B-DB) were synthesized, as described previously [27,29]. Toluene and dichloromethane (DCM) were dried over CaH_2_ and P_2_O_5_, respectively, and then freshly distilled under a N_2_ atmosphere and deoxygenated through purging with N_2_ for 30 min prior to use. Other reagents and solvents were purchased from Alfa Aesar (Haverhill, MA, USA), Aldrich Chemical Co. (St. Louis, MI, USA), Fisher Scientific (Hampton, NH, USA), Merck (Darmstadt, Germany), or Tokyo Chemical Industry Co. (Tokyo, Japan) and used without further purification. All reactions were performed under repurified N_2_ atmosphere. The monomer reaction steps are presented in Scheme 1.

### 2.2. 3′,6′-Bis(4-bromophenyl)-4′,5′-diphenyl-3,3″-bis(trifluoromethyl)-1,1′:2′,1″-terphenyl (6F7B-DB) (1)

DTF-EO (8.10 g, 11.94 mmol), diphenyl acetylene (2.55 g, 1.20 eq.) diphenyl ether (8.00 g) were placed in a 3-necked round-bottomed flask with condenser and thermometer under N_2_ atmosphere at 40 °C. The reaction temperature was then increased to 220 °C for 20 h. After being cooled to room temperature, the crude mixture was poured into methanol and deionized water (DI water), and extraction was subsequently performed using ethyl acetate and DI water; dewatering was then performed using MgSO_4_. Finally, the resultant product was obtained through reduced pressure distillation and recrystallization from tetrahydrofuran (THF)/n-hexane, producing a crystal in a 54.3% yield. ^1^H-NMR (500 MHz, CDCl_3_, 25 °C): d (ppm) 7.15 (d, 1H), 7.09 (s, 1H), 6.96–7.07 (m, 4H), 6.88–6.93 (m, 3H), 6.82–6.95 (m, 2H), 6.64–6.70 (m, 2H). Matrix-assisted laser desorption ionization–tandem time-of-flight mass spectrometry MALDI-TOF/TOF MS (*m*/*z*): Calculated for C_44_H_26_Br_2_F_6_: 828.48, found 828.30.

### 2.3. 4,4′′′′-Difluoro-3,3′′′′-bis(trifluoromethyl)-2″,3″,5″,6″,4,4′′′′-difluoro-3,3′′′′-bis(trifluoromethyl)-2″,3″,5″,6″-tetra (trifluoromethyl)phenyl-[1,1′:4′,1″:4″,1′′′:4′′′,1′′′′-quinquephenyl] (12F9B-DF) (2)

For the aforementioned reactor, 6F7B-DB (10.00 g, 12.1 mmol) and 4-fluoro-3-(trifluoromethyl) phenylboronic acid (5.00 g, 36.18 mmol) were dissolved in toluene (600 mL) under N_2_ atmosphere. Subsequently, 2.0 M K_2_CO_3_ (aq.) (5.00 g, 28.94 mmol) was added to the reactor. The mixture was boiled, (A-^ta^Phos)_2_PdCl_2_ (catalyst, 100 mg, 0.14 mmol) was added, and the mixture was then stirred for 2 days. After being cooled to room temperature, toluene and saturated saline were used to extract the organic layer. Any impurities in the mixture were removed using activated carbon, and the mixture was filtered using MgSO_4_. Finally, the crude product was purified through recrystallization from ethyl acetate/n-hexane, producing a crystal in 80.1% yield. ^1^H-NMR (500 MHz, CDCl_3_, 25 °C): d (ppm) 7.60 (d, 1H), 7.52–7.56 (m, 1H), 7.10–7.18 (m, 5H), 7.05–7.09 (m, 1H), 7.03 (d, 1 H), 6.98 (d, 1H) 6.82–6.95 (m, 7H). MALDI-TOF/TOF MS (*m*/*z*): Calculated for C_58_H_32_F_14_: 944.85, found 994.66.

### 2.4. 4,4′′′′-Dimethoxy-2″,3″,5″-triphenyl-1,1′:4′,1″:4″,1′′′:4′′′,1′′′′-quinquephenyl (8B-DMO) (3)

Briefly, 6B-DB (5.00 g, 8.11 mmol) and 4-methoxyphenylboronic acid (3.10 g, 8.11 mmol) were dissolved in toluene (800 mL) under the N_2_ atmosphere. Subsequently, 2.0 M K_2_CO_3_ (aq.) (4.50 g, 32.6 mmol) was added to the mixture. The mixture was boiled, (A-^ta^ Phos)_2_PdCl_2_ (catalyst, 100 mg, 0.14 mmol) was added, and the mixture was then stirred for 2 days. Toluene and saturated saline were used to extract the organic layer. Any impurities in the mixture were removed using activated carbon, and the mixture was filtered using MgSO_4_. Finally, the crude product was purified through recrystallization from ethyl acetate/n-hexane, producing a crystal in 85.1% yield. ^1^H-NMR (500 MHz, CDCl_3_, 25 °C): d (ppm) 7.62 (s, 1H), 7.48 (d, 2H), 7.40 (d, 2H), 7.12–7.21 (m, 9H), 7.70 (d, 2H), 6.92–6.96 (m, 5H), 6.85−6.91 (m, 9H), 6.80–6.83 (m, 2H), 3.83 (s, 3H), 3.81 (s, 3H). MALDI-TOF/TOF MS (*m*/*z*): Calculated for C_50_H_38_O_2_: 670.84, found 670.37.

### 2.5. 2″,3″,5″-Triphenyl-[1,1′:4′,1″:4″,1′′′:4′′′,1′′′′-quinquephenyl]-4,4′′′′-diol (8B-DO) (4)

All glassware used must be dried rigorously and assembled under the N_2_ atmosphere. 8B-DMO (6.00 g, 4.47 mmol) was dissolved in dry DCM (120 mL). Boron tribromide (5.1 mL, 20.4 mmol) was then injected at −78 °C under cryogenic storage dewar, and the solution was stirred for 1 day. DI water was slowly added to stop the reaction. Extraction was performed using ethyl acetate and DI water, and dewatering was performed using MgSO_4_. Finally, the resultant product was obtained through reduced pressure distillation and recrystallization from THF/n-hexane, producing a crystal in 47.0% yield. ^1^H-NMR (500 MHz, CDCl_3_, 25 °C): d (ppm) 7.62 (s, 1H), 7.43 (d, 2H), 7.34–7.37 (m, 4H), 7.21–7.12 (m, 10H), 6.93–6.95 (m, 3H), 6.85–6.90 (m, 9H), 6.80–6.83 (m, 4H). MALDI-TOF/TOF MS (*m*/*z*): Calculated for C_48_H_34_O_2_: 642.78, found 642.39.

### 2.6. 4,4′′′′-Dimethoxy-2″,3″-diphenyl-1,1′:4′,1″:4″,1′′′4′′′1′′′-quinquephenyl (7B-DMO) (5)

Briefly, 5B-DB (6.00 g, 11.1 mmol) and 4-methoxyphenylboronic acid (5.06 g, 33.3 mmol) were dissolved in toluene (700 mL) under the N_2_ atmosphere. Subsequently, 2.0 M K_2_CO_3_ (aq.) (4.50 g, 39.1 mmol) was added to the mixture. The mixture was boiled, (A-^ta^ Phos)_2_PdCl_2_ (catalyst, 100 mg, 0.14 mmol) was added, and the mixture was stirred for 2 days. Moreover, toluene and saturated saline were used to extract the organic layer. Any impurities in the mixture were removed using activated carbon, and the mixture was filtered using MgSO_4_. Finally, the crude product was purified through recrystallization from ethyl acetate/n-hexane, producing a crystal in a 63.9% yield. ^1^H-NMR (500 MHz, CDCl_3_, 25 °C): d (ppm) 7.56 (s, 1H), 7.48 (d, 2H), 7.35 (d, 2H), 7.16 (d, 2H), 6.92–6.95 (m, 5H), 6.86–6.83 (m, 2H). MALDI-TOF/TOF MS (*m*/*z*): Calculated for C_44_H_34_O_2_: 594.74, found 594.24.

### 2.7. 2″,3″-Diphenyl-[1,1′:4′,1″:4″,1′′′:4′′′,1′′′-quinquephenyl]-4,4′′′′-diol (7B-DO) (6)

As regards the aforementioned reactor and cooling bath, 7B-DOM (4.00 g, 6.73 mmol) was dissolved in DCM (150 mL). Boron tribromide (3.90 mL, 15.6 mmol) was then injected at −78 °C, and the solution was stirred for 1 day. DI water was slowly added to stop the reaction. Extraction was performed using ethyl acetate and DI water, and dewatering was performed using MgSO_4_. Finally, the resultant product was obtained through reduced pressure distillation and recrystallization from THF/n-hexane, producing a crystal in 92.1% yield. ^1^H-NMR (500 MHz, CDCl_3_, 25 °C): d (ppm) 7.56 (s, 1H), 7.44 (d, 2H), 7.34 (d, 2H), 7.16 (t, 2H), 6.92–6.96 (m, 3H), 6.83–6.88 (m, 4H). MALDI-TOF/TOF MS (*m*/*z*): Calculated for C_42_H_30_O_2_: 566.69, found 566.22.

### 2.8. General Procedure for the Synthesis of Polymers

Polymerization reactions were conducted in a three-necked 100-mL flame-dried flask equipped with a stirring bar and Dean–Stark apparatus fitted with a condenser under N_2_ atmosphere. The flask was charged with K_2_CO_3_ (3.61 mmol), dimethylacetamide (DMAC; 25 mL), dry toluene (15 mL), difluoro monomer 12F9B-DF (1.50 g, 1.51 mmol), and bisphenol monomer (1.51 mmol). Three types of polymers were used: 9B-DO, 8B-DO, and 7B-DO. The polymer reaction steps are presented in Scheme 2.

The one-pot reaction was dried through azeotropic distillation at 130 °C for 2 h; then, the reaction temperature was increased to 160 °C for 36 h. After being cooled to room temperature, the crude mixture was poured into methanol and DI water to precipitate a beige fibrous polymer. The polymer was filtered, washed several times with water and hot methanol, and dried in a vacuum at 80 °C for 8 h.

### 2.9. General Procedure for Sulfonation

To a solution of the polymer (1.20 g) in DCM (75 mL) at room temperature, chlorosulfuric acid in DCM was added dropwise. The reaction mixture was stirred for 24 h and then poured into water. The polymer precipitate was filtered, washed thoroughly with DI water until the pH became neutral, and then dried in a vacuum at 60 °C overnight to obtain the sulfonated polymer. The polymers were sulfonated to different extents by using the aforementioned procedure and adding 4 or 12 mL of a sulfonating agent, respectively. The sulfonated polymer was readily soluble in polar aprotic solvents such as dimethylformamide, DMAC, DMSO, and n-methyl-2-pyrrolidone (NMP). The ionomer reaction steps are presented in Scheme 2.

### 2.10. Measurements

^1^H NMR spectroscopy was performed using a 500-MHz Varian Unity Inova 500 spectrometer using CDCl_3_ or DMSO-d6 as the solvent. MALDI-TOF mass spectra were obtained using a Bruker Daltonics Autoflex III TOF/TOF with 2,5-dihydroxybenzoic acid employed as the matrix. The FTIR spectra of the polymer membranes were obtained using a Bruker VERTEX 70 FTIR spectrometer. Gel permeation chromatography analysis was conducted using a Viscotek 270 Max with a refractive index detector and THF used as the eluent at a flow rate of 1 mL min^−1^. For calibration, a polystyrene standard (molecular weight between 75 and 117 kDa) was used. Stress–strain curves were obtained for the film specimens (length, 10.0 mm; width, 1.0 mm; and thickness, 30–40 μm) by using the PerkinElmer Pyris Diamond TMA at 25 °C. Membranes with various thicknesses may possess different physical–chemical properties, including mechanical strength, proton conductance (not electrical conductivity), water uptake, and resistance [34]. Accordingly, the thickness was controlled between 30 and 40 μm to match the comparable range of the selected standard, Nafion 211 (25.4 μm). For sufficient ion cluster growth time, it is necessary to have both high-concentration dissolution characteristics and appropriate solvent evaporation rate under film-forming temperature conditions. Therefore, this study uses high-boiling DMSO as the casting solvent.

### 2.11. Stability

The thermal stability of the polymers was evaluated through thermogravimetric analysis (TGA) performed using a PerkinElmer Pyris 1 from 50 to 800 °C at a heating rate of 10 °C min^−1^ under N_2_ atmosphere. Before the analysis, the membranes were dried in the TGA furnace at 130 °C under N_2_ for 1 h to remove moisture.

The membranes were weighed and soaked in Fenton’s reagent (3% H_2_O_2_ aqueous solution containing ferrous sulfate at 2.0 ppm) at 80 °C for observation after 24 h. Oxidative stability (OS) was evaluated using the change in weight of the membranes after exposure to Fenton’s reagent.

### 2.12. Water Uptake and Dimensional Stability

The water uptake (WU) of the membranes is gauged by comparing the weights of the dry and the wet membrane samples by means of Equation (1). The dry membrane weight (*W_dry_*) is obtained by vacuum drying the sample at 80 °C for 24 h immediately before weighing it. The weight of the corresponding membrane in wet conditions (*W_wet_*) is obtained by immersing the membrane sample in DI water at a specified temperature for 24 h, wiping off the surface moisture with filter paper, and then quickly weighing it. The final *WU* is obtained from the average of three experiments.
(1)$$Water\;uptake\;\left(WU\right)=\frac{W_{wet}-W_{dry}}{W_{dry}}\times 100\%,$$

The swelling ratio was calculated using the following Equations (2) and (3):(2)$$In\;plane\;selling\;ratio\;\left(\Delta T\%\right)=\frac{T_{wet}-T_{dry}}{T_{dry}}\times 100\%$$
(3)$$Through\;plane\;selling\;ratio\;\left(\Delta A\%\right)=\frac{A_{wet}-A_{dry}}{A_{dry}}\times 100\%$$
where *A_wet_* and *A_dry_* are the areas, and *T_wet_* and *T_dry_* are the thickness of the wet and dry membranes, respectively. 

### 2.13. Ion Exchange Capacity

The ion exchange capacity (IEC) of the membranes was determined through acid-base titration. Each dried membrane was weighed and then immersed in 1.0 M HCl (aq.) for 24 h to protonate the acid groups; it was then washed thoroughly with DI water to reach a neutral pH. Subsequently, the membrane was immersed in a 1.0 M NaCl (aq.) for 24 h to replace the protons of the sulfonic acid groups with Na+ ions. The solution was titrated using 0.01 M NaOH (aq.), and phenolphthalein was used as an indicator. The IEC (mmol g^−1^) was calculated from the titration results by using the following Equation (4):IEC (mmol g^−1^) = (*V_NaOH_* × *M_NaOH_*)/*W_dry_*(4)
where *V_NaOH_* and *M_NaOH_* are the volume and concentration of the NaOH solution, respectively, and *W*_dry_ is the weight of the dry membrane.

### 2.14. Hydrated Molar Volume per Charge

The molar volume per charge (*MVC*) is an estimate of equivalent volume (cm^3^ per ionomer or the mol equivalent of acid groups) based on the summation of molar volume subunits rather than true volume measurements, as shown in Equation (5) [31,35] as follows:(5)$$MVC=\sum_{i}n_{i}V_{i}$$
where *V_i_* is the volumetric contribution of structural group *i*, which appears *n_i_* times per charge.

The hydrated molar volume per charge (*MVC*_(*wet*)_) was the sum of the molar volumes of the component atoms or water uptake of the polymers, which is given by Equation (6) [31,35] as follows:(6)$$MVC_{\left(wet\right)}=\sum_{i}n_{i}V_{i}+V_{H_{2}O}\times \lambda$$
where $V_{H_{2}O}$ is the molar volume of water, 18 (cm^3^ mol^−1^)., and *λ* is the number of water molecules per charge based, as shown in Equation (7) as follows:(7)$$\lambda =\frac{\left[H_{2}O\right]}{\left[SO_{3}^{-}\right]}=\frac{10\times WU}{M_{H_{2}O}\times IEC}$$
where $M_{H_{2}O}$ is the molar mass of water.

### 2.15. Percent Conducting Volume

Percent conducting volume (*PCV*), which is the ratio of the water uptake volume to the acid volume concentration, was calculated using the following Equation (8) [31,35]:(8)$$PCV=\frac{V_{H_{2}O\;}\times \lambda }{MVC_{\left(wet\right)}}$$

Recently, research teams have employed this equation to better compare the proton conductivity of sulfonated polymers in PEMs in various states of WU [31,35,36].

### 2.16. Proton Conductivity

The membranes’ proton conductivity was measured using a frequency response analyzer (Solartrom 1260) along the in-plane direction over the frequency range 10 MHz to 100 Hz at a voltage amplitude of 100 mV. Conductivity was measured after clamping a 10.0 mm × 5.0 mm sample between the two platinum electrodes of a conducting cell. The test cell was placed in an Espec SH-241 environmental chamber to measure its conductivity at 80 °C and various values of relative humidity (RH). Proton conductivity (σ) was then calculated using the equation σ = *L*/*RA*, where *L* (cm) is the distance between the electrodes, *R* (Ω) is the membrane’s resistance, and *A* (cm^2^) is the cross-sectional area of the sample.

### 2.17. Microstructure Analysis

Transmission electron microscopy (TEM) was performed using a JEM-2100 (high-resolution) TEM instrument operated at an accelerating voltage of 200 kV. The acid form of the s-P12FmB-X membranes was dyed and converted into Ag^+^ ions through overnight immersion in 1.0 M AgNO_3_. Subsequently, the membranes were washed thoroughly with DI water and vacuum dried at 80 °C for 12 h. These dyed membranes were placed in an enclosure of epoxy resin and ultramicrotome under cryogenic conditions to obtain samples with a thickness of 30 nm.

### 2.18. Single-Cell Performance

Catalyst ink was prepared by mixing Pt/C (HiSPEC 4000, Alfa Aesar, Haverhill, MA, USA) with a 5 wt% Nafion D520 binder. The ink was sprayed onto both sides of the membranes. The active surface area was 5 cm^2^ with an overall Pt loading of 0.4 mg cm^−2^. A membrane electrode assembly (MEA) was obtained by sandwiching the s-P12FmB-X membranes between two gas diffusion layers (GDL 24 BC, Hephas Energy Co., Ltd., Hsinchu, Taiwan). An MEA featuring a Nafion 211 membrane was fabricated using the same procedure to serve as a reference. The anode and cathode were supplied with hydrogen at a flow rate of 0.2 L min^−1^ and oxygen at a flow rate of 0.4 L min^−1^, respectively. The fabricated cell was activated for 4 h with hydrogen and oxygen at 80 °C and 0.5 V.

## 3. Results and Discussion

### 3.1. Synthesis and Characterization of the Monomers and Polymers

A novel tetra-trifluoromethyl-substituted hexaarylbenzene (HAB) derivative difluoro monomer was successfully synthesized in three steps, as illustrated in Scheme 1. Using the method discussed in our previous report, DTF-EO was synthesized from 1,3-bis(4-bromophenyl)propan-2-one and 1,2-bis(3-(trifluoromethyl)phenyl)ethane-1,2-dione through aldol condensation. DTF-EO was then reacted with diphenylacetylene through the Diels–Alder reaction to obtain monomer1 (6F7B-DB). To obtain the leaving for the condensation reaction, monomer 2 (12F9B-DF) was prepared from 6F7B-DB and 4-fluoro-3-(trifluoromethyl)phenyl)boronic acid through Suzuki coupling. To obtain the diol monomers, a series of volume-related MAB guides—nB-DB, where m is the number of Benzene Rings and synthesized, as described previously. Then, nB-DMO series were synthesized from three guides and 4-methoxyphenylboronic acid through Suzuki coupling. Finally, diol monomers (nB-DO) were prepared from nB-DMOs through BBr_3_ demethylation. All the monomer structures were confirmed through ^1^H NMR and MALDI-TOF-MS spectroscopy (Appendix A).

The polymers were synthesized through S_N_Ar with a difluoro monomer (12F9B-DF) and diol (nB-DO); these polymers are denoted P12F99B, P12F98, and P12F97B, respectively, as shown in Scheme 2. The polymers were readily soluble in common solvents DMAC, DMSO, NMP, THF, and chloroform. The P12FmB series polymers had molecular weights of 75–117 kDa and polydispersity indices of 2.0–2.7 mmol g^−1^_._ These results reveal that a short perfluoroalkyl group could contribute sufficient solubility, and the tetra-trifluoromethyl substitution in the phenyl provided a strong electron-withdrawing group to increase the substitution activity of halophenyl enough to obtain high-molar-mass poly(arylene ether)s during polymerization [37,38].

Postsulfonation of the P12FmB polymers was conducted using chlorosulfonic acids of various concentrations to obtain s-P12FmB-X ionomers, where m indicates the composition of phenyl rings and X indicates the IEC. The structures were confirmed through ^1^H NMR spectroscopy. The NMR signals of ionomers and polymers are presented in Figure 1, Appendix A, respectively. After polycondensation, the NMR signal of the P12FmB series polymers comprised a single sharp peak (labeled 12) and was shifted to 7.70 ppm because of a change in the chemical environment of the trifluoromethyl aromatic ring at the junction. For instance, inP12F98B, as trifluoromethyl passivates the strong electron-withdrawing and deactivating aromatic ring, the substituent of the sulfonated group is often not in the trifluoromethyl substituted phenyl group. The new signal at 7.59–7.77 ppm is assigned to the proton next to the sulfonated pendant benzene ring; the chemical shift is similar to that reported in our previous work [27,28,39].

To confirm the sulfonation of the polymers, FTIR spectroscopy was performed to determine the structural composition of the untreated polymer and polymers with different degrees of sulfonation. After the sulfonation treatment, a broad absorption band was generated from 3749 to 2425 cm^−1^ (–OH stretching) and was accompanied by changes in the absorption peaks in fingerprint regions, as shown in Appendix A, which reveals a slight shift of the peak corresponding to symmetrical stretching of C=C absorption from 1492 to 1478 cm^−1^. Larger absorption peaks appear at 1350 and 1010 cm^−1^ and correspond to symmetrical stretching and in-plane bending of the O=S=O functional group [40,41,42]. The appearance of an enlarged shoulder at 1225 cm^−1^ is ascribed to asymmetric stretching [43]. The peak corresponding to the absorption of aromatic ethers (Ar–O–Ar) on the main chain remains at 1035 cm^−1^ [42,44].

### 3.2. Thermal Stability

The s-P12FmB-X series arylene ether ionomers obtained through sulfonation were dried to remove moisture and then kept at 120 °C for 30 min before their thermal stability was evaluated to guarantee the high stability of the cell during operation. Compared with the untreated polymers, the s-P12FmB-X series ionomers exhibited less than 2 wt% thermogravimetric loss due to loss of residual moisture before 120 °C. The trend, similar for all of the sulfonated polymers, consisted of a drying stage and two degradation stages. The thermal degradation curve obtained at 200–400 °C was attributed to the decomposition of sulfonic acid groups [45]. The secondary degradation curve obtained above 600 °C approximately corresponded to the degradation of the polymers’ main chain. Figure 2 presents the degradation of polymers, and their 5% weight loss temperature (*T*_d5%_) is listed in Table 1. All the untreated polymers exhibited excellent thermal stability, with *T*_d5%_ higher than 589 °C. This was attributed to the multiple phenyl ionomer because rigid aromatic groups are known to provide strong resistance to thermal degradation [14].

### 3.3. Mechanical Properties

The mechanical properties of the P12FmB (innate form, solid symbol lines in Figure 3) and s-P12FmB-X (acid form, hollow symbol lines in Figure 3) series membranes and of the control sample (PFSA, Nafion 211) were measured; the detailed results are shown in Table 1. The innate-form membranes were discovered to have Young’s moduli of 1.60–1.83 GPa, a tensile strength of 80.3–97.1 MPa, elongation at break of 25.6–40.3%, and excellent source characteristics. The toughness of poly(arylene ether)s is derived from the contribution of rigid aromatic groups and their van der Waals forces, giving the full range of polymers similar strength and generally low elongation at break as a more brittle characteristic. The elongation at break P12F97B > P12F98B > P12F97B can be attributed to the architecture effect of the polymer chains. P12F97B has fewer side chains on the main chain, which allows it to stack better and obtain slightly stronger properties. After treatment, the degree of acidification was higher, and the strength was lower, conforming to the characteristics of strongly hydrophilic group implantation [8,46,47]. Moisture acted similar to a plasticizer to enhance the ductility of the membranes (elongation at break, 57.7–95.8%) but combined with the rigid structure of the poly(arylene ether)s to yield sufficient toughness. Although a significant increase in elongation at break was obtained after the sulfonation treatment, a decrease in strength was observed in Table 2. The sustained increase in IEC does not allow for better toughness of the membranes. Consequently, Young’s moduli and tensile strength of the acid-form membranes were 0.34–0.57 GPa and 48.7 MPa–82.8 MPa, respectively. This result demonstrated that the mechanical properties of the s-P12F97B-X series of membranes were comparable to those of Nafion 211 membranes (Young’s modulus, 0.28 GPa; tensile strength, 34.2 MPa). Regarding the elongation at break, s-P12F97B-2.92 > s-P12F98B-3.00 > s-P12F97B-2.89, which is attributed to the following two points: 1. the enhancement of IEC is accompanied by the enhancement of WU, and the effect of self-wetting increases the plasticizer content and enhances the ductility; 2. the stacking surface, which mainly contributes to van der Waals forces, changes with the substitution of sulfonic acid groups, decreasing van der Waals forces. Since the active zone of sulfonation substitution (diol-monomers, Scheme 2) is also located at the difference of the three polymer structures, the increase of IEC also means that the force dominance changes from van der Waals forces to hydrogen bonding. In the presence of hydrogen bonding and plasticizer (moisture), the ductility is increased and the strength is decreased, allowing for proper toughness. The detailed hydration behavior will be discussed subsequently.

### 3.4. Hydration Behavior

Fuel cells are operated under high humidity, and the water management characteristics of PEMs must not be neglected. Excellent water uptake (WU) for the proton carrier may be necessary, and its powerful influence on fuel cell performance [2,9,48]. However, the hygroscopic swelling of polymers is accompanied by dimensional changes, microphase transitions, and even polymer corrosion [9]. The WU (Figure 4), hydration number, dimensional stability, and oxidative stability (OS) of the s-P12FmB series membranes were investigated, and the results are listed in Table 2. The Ave designation in the s-P12FmB-Ave implies the average characterization of this series of ionomers. Since the IEC represented by X cannot be mathematically averaged to express its significance, the average characterization and distribution are expressed as s-P12FmB-Ave.

As expected, the WU of the membranes increased with an increase in temperature, IEC, and the number of arms of the phenyl group substituent. The amount of WU of the s-P12FmB series membranes was 24–67% at 30 °C but increased to 74–143% at 80 °C. The highest WU, for the s-P12F99B-2.89 membrane, was twice that of the others at 80 °C. In addition, the slightly higher IEC of the polymer was attributed to the numerous sulfonated active sites and larger free volume on the structure. Within the range of comparable IECs, s-P12F98B-3.00, which had an asymmetric notch in its structure, had the highest initial WU (60%) at low temperatures, and this uptake was greater than that of Nafion 211. The effective WU properties are expected to provide an increase in the proton conductivity of the film. Typically, water is required and generated during the operation of a PEM fuel cell. As a result, membranes with excessive swelling rates may shrink under repeated cell on/off (charge/discharge cycling) operations, resulting in creep and the risk of catalytic layer peeling. In this study, the drawbacks of obtaining films with acceptable swelling rates need to be verified by subsequent rigorous fuel cell lifetime studies.

In this system, the hydration number (*λ*), the number of equivalent carriers that participate in carrying ions that are calculated using the experimental value of WU, was equivalent to the number of water molecules per sulfonic acid group. As the result, the s-P12F99B-X series membranes had higher *λ* (average 27.0) under operating conditions (80 °C; RH, ~99%). This may be because the free space and the distribution of sulfonic acids in the structure are relatively concentrated. On the other hand, the *λ* of the s-P12F98B-Ave and s-P12F97B-Ave membrane were approximately 19.0 and 14.8, respectively; the *λ* value decreases as the number of arms decreases. In this case, the equivalent carrying capacity of the s-P12F99B-Ave membrane may be better. Under similar IEC conditions, *λ* was discovered to increase with an increase in the number of sulfonyl phenyl arms, indicating that the densely hydrophilic concentration affects the hydration behavior.

The swelling of the s-P12FmB series membranes resulted in an approximate 30–120% change in area and 30–60% change in thickness at 80 °C, as illustrated in Figure 5. Notably, the s-P12F97B series membranes exhibited excellent dimensional stability, with an average area change of 43.8 ± 10.3% and thickness change of 42.4 ± 3.6%. The s-P12F99B-3.23 membrane had the largest dimensional change at low temperature because of its lowest trifluoromethyl content and highest water absorption capacity, which were crushing at 80 °C. This result can be attributed to the variable free volume and rigidity of the multiphenylated structure; the sulfonated territory allows water to occupy it [49,50], and the strong hydrophobic phase (–CF_3_) improves the solubility and acceptable swelling rate of the membrane [27,51].

The resistance of the membranes to oxidation was tested using Fenton’s reagent, and all membranes exhibited excellent OS at 80 °C for 1 h; the results are listed in Table 2. After 16 h, continuous swelling was observed, and only the s-P12F99B-3.23 membranes completely degraded. The degradation of the s-P12F99B-Ave film was attributed to its high WU, which caused collapse and dissolution and increased the reaction area. Until 24 h, the asymmetric s-P12F98B-Ave still had higher oxidative stability than s-P12F97B-Ave—58.6 and 52.5 wt%, respectively. Inevitably, the long-term stability of the prepared membranes is not that optimistic or promising, compared to Nafion membranes. It is recommended to introduce functional groups that can effectively form electron-deficient structures in the polymer structure to improve the OS of membranes [37]. Although the trifluoromethyl group can effectively form an electronic trap structure, the OS was expected to increase as the volume ratio of trifluoromethyl molecules in the repeating unit was increased [52,53]. Up to 24 h, the stability of the s-P12F98B-Ave was similar to that of the s-P12F97B-Ave—58.6 and 59.5 wt%, respectively. This implies a threshold degree of intensive sulfonation as a function of the number of arms of phenyl group substituents, and this threshold can be used as a design reference for subsequent research.

### 3.5. Proton Conductivity

Proton conductivity is a principal indicator in the evaluation of the mass transfer characteristics of an MEA, which usually involve multiple transport mechanisms and hydration behaviors on multiple scales [9,26,32,37,54,55]. This article describes space-related hydration factors such as IEC, *λ*, MVC_(wet)_, and PCV; the characteristics are summarized in Table 2. The proton conductivity of the s-P12FmB series membranes was measured as a function of RH at 80 °C and compared with that of the iconic standard Nafion membrane (Figure 6). Compared with the Nafion membrane (123.8 mS cm^−1^) under the same measurement conditions, the s-P12FmB membranes (>173.9 mS cm^−1^) had superior proton conductivity due to the dense sulfonated structure. The conductivity of the s-P12FmB membranes decreased with a decrease in the RH but was higher than that of the Nafion at all RHs. The proton conductivity of the s-P12FmB-Ave membranes with comparable IECs increased gradually with an increase in the number of phenyl groups in the structure; this increase may be attributed to the higher concentration of local sulfonate groups. Due to the existence of trifluoromethyl groups, the probability of sulfonation of the benzene ring can be reduced [13], and the concentration of localized sulfonate groups on the polymer chain can be further increased. Accordingly, s-P12F99B-2.89 had high proton conductivity (294.1 mS cm^−1^) at low IEC, whereas s-P12F98B-3.00 and s-P12F97B-2.92 had lower proton conductivity (219.0 and 186.6 mS cm^−1^, respectively) at high IEC.

Due to the nature of the monomer components, the molecular volumes of their repeat units were different, and the IEC was introduced; the MVC values of the polymers in the dry state were calculated. To approximate the operating conditions of a fuel cell, the *λ* under high humidity and temperature was measured to obtain the MVC_(wet)_ in this case. Although the actual volume was not obtained, when saturated, the estimated volume change was assumed to be mainly due to the filling effect caused by the ion clusters absorbing water and swelling under high humidity [9,31,36,52]. Incidentally, we can notice that the conductivity of s-P12F97B-X membranes under low humidity conditions has a contradictory behavior. It does not comply with the rising trend of conductivity with the increase of IEC. However, the results from MVC to MVC_(wet)_ can be observed to be very similar, which means that the equivalent space provided by the concentration of sulfonate [–SO_3_H] is similar. With the increase of RH, the master control of conductivity returns to IEC and returns to the normal trend at high RH conditions. Accordingly, we believe that the counterintuitive behavior of s-P12F97B-X membranes under low humidity conditions had an acceptable margin of error.

To determine the influence of hydration relative to conductivity, the influence of free volume was focused on the hydration domain (water-rich areas in the membrane), and the hydration ratio of ionomers under fully hydrated conditions was determined as the PCV; the result is listed in Table 3. With the change in the number of side-chain arms, the difference in IEC was insufficient to demonstrate the effect of the hydration zone on conductivity, whereas the PCV clearly differed. Consequently, even if the IEC was different, the PCV exhibited by ionomers with the same number of arms in the system under full hydration was similar. For example, even if the *λ* value of the s-P12F99 membranes was different, the proportion of domains that were hydrated was similar (PCV ≈ 0.71), and the conductivity was also similar. Conversely, as the PCV decreased, the conductivity decreased considerably, such as for s-P12F98B-2.61 and s-P12F98B-3.00. Although s-P12F98B-2.61 had a higher MCV_(wet)_ (562.8 and 543.8 cm^2^ [eq. mol.]^−1^, respectively), its PCV was low (0.60 and 0.63, respectively); the ratio of the volume of the hydrophilic phase to that of the hydrated membrane was relatively small; therefore, the conductivity was low (255.1 and 212.6 cm^2^ [eq. mol]^−1^, respectively). s-P12FmB series membranes decreased PCV with an increase in the number of arms that were sulfonated. Notably, the membrane had low sensitivity to humidity, and the conductivity remained higher than 100 cm^2^ (eq. mol)^−1^ and was comparable with the Nafion membrane (39–65 cm^2^ [eq. mol]^−1^) at low humidity (RH, 60–80%). The low-humidity sensitivity is beneficial because humidity often varies during cell operation.

### 3.6. Microstructure Analysis

The s-P12FmB series membrane had clear phase separation, including main isolated clusters (size, 3–8 nm) and mesoscale leopard-like clusters (15–45 nm), as visualized in the dry state by using TEM (Figure 7). In Figure 7, the shadowy areas represent hydrophilic ionic clusters, whereas the brighter areas represent the hydrophobic polymer matrix. The expansion of the shadowy area implies the expansion of the proton transmission channel. The aforementioned characteristics are observable to varying degrees in the cross-sectional TEM images of the s-P12FmB series membranes. The overall shape of the s-P12F99B-2.89 membrane was gyroid. In the microphase separation, isolated clusters tended to be connected to condensate subregions with low sulfonate density and a diameter of approximately 45 nm, as shown in Figure 7a. In the image of the s-P12F98B-3.00 membrane, the tendency to connect appears to be reduced, forming a metastable state between the gyroid and cylindrical types, as shown in Figure 7e. Under similar IEC, the number of side chains passing through the aromatic group decreased, and the density and morphology of large clusters changed until round isolated clusters formed in the s-P12F97B-2.92 membrane, as revealed in Figure 7f. Thus, the trend in the microphase separation pattern could be successfully changed by controlling the number of aromatic side chains. Predictably, the generated operating groups produced during the cell process form the main path for further growth while being transported.

### 3.7. Fuel Cell Performance

A single-MEA fuel cell with suitable mechanical characteristics and excellent proton conductivity was fabricated using s-P12FmB-X series and Nafion 211 membranes with overall Pt loading of 0.4 mg cm^−2^ on both sides of the membrane. The fuel cell was examined at 80 °C at full hydration. Figure 8 shows the polarization and power density curves of the MEA containing the s-P12FmB-X and Nafion 211 membranes. The s-P12F97B-2.92 cell exhibited an excellent current density of more than 1670 mA cm^−2^ (at 0.6 V) and power density similar to that of the MEA containing Nafion 211 with identical graphite bipolar plate and other fuel cell operating conditions. As shown in Figure 8, the cell with the s-P12F97B-2.92 membrane demonstrated the highest maximum power density (1.32 W cm^−2^) and had superior power density to that containing Nafion 211. The fuel cell containing the s-P12F98B-3.00 membrane performed well (current density > 1650 mA cm^−2^ at 0.6 V), but greater ohmic region loss induced power density that was inferior to that for Nafion 211 (>2000 mA cm^−2^). On the other hand, all s-P12FmB-X series membranes are known to have reached the realm of overhydration (PCV > 0.35) from PCV, which means that the hydrophilic phase may extend to satiation. The original TEM images used to evaluate the ion channels will need to be discussed in terms of their opposites. The gyroid from the ion cluster, which was expected to be beneficial for proton transport, instead has a detrimental effect on the structural support of the membrane. In addition to the macroscopic swelling of the membrane under high temperature and high RH conditions during cell operation, water crossover and fuel permeation may also occur. As a result, the s-P12F99B-2.89 membrane, which has the highest proton conductivity characteristics, exhibited cells performance that were not that expected. The optimization of membrane permeation modulation IEC may be the direction of research to obtain improved cell performance. Encouragingly, this study has been successful in improving cell performance through a slight structural design (different number of phenyl substitution arms and dense sulfonation), which is also expected to provide a reference for subsequent researchers. Accordingly, the high performance of the cell containing the s-P12F97B-2.92 membrane may have been due to its high dimensional stability and sufficient proton conductivity.

## 4. Conclusions

In summary, we introduced a series of novel sulfonated poly(arylene ether)s that contain superhydrophobic tetra-trifluoromethyl-substituted HAB and three types of densely sulfonated phenylated MAB structures. The polymers were prepared through nucleophilic substitution polycondensation, and postsulfonation was performed using chlorosulfuric acid. The electron-withdrawing pendant groups of trifluoromethyl activated polymerization and improved solubility to yield high-molecular-weight polymers. The results indicate that trifluoromethyl inhibits the sulfonation activity of local phenyl substitutions and postsulfonation can achieve more precise positioning in polymer while maintaining the distribution of hydrophobic and hydrophilic moieties. Furthermore, all of the membranes exhibited high thermal stability, favorable dimensional stability, and conductivity that was not sensitive to humidity. The fuel cell containing the s-P12F97B-2.92 membrane demonstrated excellent performance with a current density of more than 1670 mA cm^−2^ at 0.6 V and a maximum power density of 1.32 W cm^2^ at 80 °C at full hydration, properties that are superior to those of Nafion 211. In addition, the s-P12F97B membranes exhibited tensile stress at a maximum load of 59.6–76.6 MPa and elongation at break of 82.8–95.8%. The combination of high thermal stability, acceptable dimensional stability, high proton conductivity, and excellent single-cell performance makes s-P12F97B-2.92 attractive as a PEM material for fuel cell applications.

## Data Availability

Data are available in a publicly accessible repository.

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
