# Peer review of "Highly Proton-Conducting Membranes Based on Poly(arylene ether)s with Densely Sulfonated and Partially Fluorinated Multiphenyl for Fuel Cell Applications"

_membranes, 2021, doi:10.3390/membranes11080626_

Round 1
Reviewer 1 Report
The paper deals with synthesis of series of partially fluorinated sulfonated poly(arylene ether)s as proton exchange membranes for fuel cell application. The authors found that the s-P12F97B membranes is a good candidate for proton exchange membranes. The authors did a lot of works on synthesis and characterization of membranes. However, this work mainly focuses on the description of the behavior of membranes with very poor explanation.
The authors should explain why the stress of P12F97B, P12F98B, P12F99B are similar while the stress of s-P12F97B-2.92 is strongly higher than that of s-P12F98B-3.00 and s-P12F98B-3.23 (Figure 2).
Similarly, the elongation at break of membrane looks like increasing after sulfonation. Why are the elongations at break of the lower sulfonated membranes such as s-P12F99B-2.89, s-P12F98B-2.61, s-P12F97B-2.84 higher than s-P12F99B-3.23, s-P12F98B-3.00, s-P12F97B-2.92, respectively (table1)?
The proton conductivities of the s-P12F99B and s-P12F98B membranes look like increasing with IEC. Why do we observe the contradictory trend in s-P12F9B membranes?
The authors said that the high dimensional stability and sufficient proton conductivity of the membrane may be the reason of the high performance of the fuel cell. It seems not be a satisfied explanation because the stability and proton conductivity of s-P12F99B-3.23, s-P12F98B-3.00 are also higher than Nafion 211 even the performances of the fuel cells containing them are lower than that containing Nafion 211. The author should show the performances of the fuel cells containing other membranes.
The authors should mention more detail of synthesis of monomers from 1 to 6.
The authors should show exactly used membranes in figures 3, 4, 5, 7.
I can not see the clear relation of characterization of membranes to the performances of the fuel cells.
Reviewer 2 Report
Overall review comments
The manuscript reported a series of partially fluorinated sulfonated poly(arylene ether)s through nucleophilic substitution polycondensation with post-sulfonation to obtain densely sulfonated ionomers. The membranes exhibited considerable dimensional stability and oxidative stability. The proton conductivity of the membranes is higher than that of Nafion membranes. A fuel cell test containing one such material exhibited high single-cell performance as well. The results indicate that the material is a candidate for PEM in fuel cell applications.
I consider the content of this manuscript will definitely meet the reading interests of the readers of the Membranes journal. However, the membrane thickness is not discussed at all, and the influence of various solvents is missing. The descriptions have certain issues/drawbacks, which is very confusing to the readers.
Therefore, I suggest giving a minor revision and the authors need to clarify some issues or supply some more experimental data.
Principal comments
- In the experimental part, I strongly recommend that the authors add a chemical reaction process scheme/chemical reaction flow chart to make the reader more clear about the entire reaction process (not in the discussion part, but directly in the experimental part). The text description is too long and monotonous, and it is easy to get lost in the process of
- The thickness of the obtained membrane should be demonstrated directly in the text or in some of the Membranes with various thicknesses may possess different physical-chemical properties, including mechanical strength, proton conductance (not conductivity), water uptake, and resistance [International Journal of Energy Research 43.14 (2019): 8739-8752]. And also, if only one thickness of the samples is selected, the authors should explain why it is specially selected.
- Line 458, ‘Up to 24 h, the stability of the s-P12F98B film was similar to that of the s-P12F97B film—58.6 and 5 wt%, respectively.’ However, it should be noted that the OS value of Nafion 211 is as high as 88.6%. Therefore, the long-term stability (one month or even more) of the prepared membranes is not that optimistic or promising compared to Nafion membranes. Is this a drawback for the obtained membranes, and how can further improve it? More discussion is needed for such a result.
- In this work, only dimethyl sulfoxide (DMSO) is adopted as the solvent for solution casting. Is there some special reason for selecting this solvent? Since various solvents during the casting process may have a great influence on the final physical-chemical properties of the final membrane result, the authors should clarify this issue or adopt various solvents for such membrane investigation [Journal of Power Sources 285 (2015): 195-204].
Secondary/Minor issues
1. Introduction
Line 20, ‘The proton conductivity of the membranes, higher (174.3 – 301.8 mS cm− 1) than that of Nafion 117 (123.8 mS cm − 1)’. Here the conductivity is Nafion 117, while in the latter Table, the conductivity of Nafion comes from Nafion 211. The authors need to confirm which Nafion membranes they select and be identical about the description before and after.
Line 30 to Line 36, during the introduction of fuel cells, it should be mentioned about the origin of hydrogen energy (from water splitting). And fuel cells can be used in combination with water electrolysis devices to store intermittent renewable energy. In addition, for energy storage applications, what is the difference or advantages of fuel cells in comparison with lithium-ion battery [Advanced Energy Materials 10.18 (2020): 1904152] and redox flow battery [Journal of Power Sources 493 (2021): 229445] should also be discussed briefly.
Line 63, ‘they can be adjusted structural design of polymer composites to improve by the ion exchange capacity and microphase separation [24]’. What is ‘microphase separation’? I consider it may be the phase separation between hydrophobic domains and hydrophilic domains, which provides the proton transportation pathways. The authors need to further clarify this issue by more details here.
2. Materials and Methods
Throughout the experiment, too many chemical reagent names are very lengthy, making readers feel boring. Is it possible to consider replacing certain reagents with appropriate abbreviations to make the description of the experimental process easier?
The NMR results do not seem to be displayed in the experimental part, the structure is slightly confusing, and the preparation process and results are shown together.
Line 231 to Line 234, the weight of dry and wet membrane is not defined clearly. A more detailed example can be, ‘The water uptake (WU%) of the membranes is gauged by comparing the weights of the dry and the wet membrane samples by means of Eq. (1). The dry membrane weight (Wdry) is obtained by drying the sample at 100 °C for 12 h immediately before weighing it. The weight of the corresponding membrane in wet conditions (Wdry) is obtained by immersing the membrane sample in bidistilled water for 24 h, wiping off the surface moisture with filter paper and then quickly weighing it. The final water uptake is obtained from the average of three experiments.’ [Solid State Ionics 319 (2018): 110-116; Rsc Advances 6.5 (2016): 3756-3763]. The same applies to the calculation of the ‘swelling ratio’.
Line 296, ‘The anode and cathode were supplied with hydrogen at a flow rate of 0.2 L min−1 and oxygen at a flow rate of 0.4 L min−1, respectively.’ Why is the flow ratio between hydrogen and oxygen is 1:2, not 2:1?
3. Results and Discussion
About the FTIR assignment, 1492 to 1478 cm − 1 should be related to the aromatic ring symmetrical stretching of [C=C]. 1010 cm − 1 may also be stretching of [C-O-C], and 1250 cm-1 be symmetrical stretching of [O=S=O], as reported in [Table 5, Electrochimica Acta, 309, 311-325]. The authors need to comprehensively consider the possibility of peak assignments.
Line 417, ‘Within the range of comparable IECs, s-P12F98B- 3.00, which had an asymmetric notch in its structure, had the highest initial WU (60%) at low temperatures, and this uptake was greater than that of Nafion 211.’ Here, it is not to say that only s-P12F98B-3.00 with the highest initial WU is higher than that of Nafion 211. Indeed, all the prepared samples have much higher WU than Nafion 211. In addition, with the temperature increasing, the WU of Nafion 212 is quite stable. But the WUs of the prepared samples increase a lot, which also indicates a very significant swelling phenomenon. Will excessive swelling rate adversely affect the actual application of the membrane materials described in this manuscript? Appropriate discussion should be added here.
Line 542, ‘The s-P12F97B-2.92 cell exhibited excellent current density of more than 1670 mA cm−2 (at 0.6 V)’. I do not think so, it seems the current density of s-P12F97B-2.92 (green line) is lower than 500 mA cm-2 at 0.6V.
In Figure 7, why the curve colour of s-P12F99B/s-P12F97B is not identical for the voltage and power density? It is very confusing for the readers.
Round 2
Reviewer 1 Report
The manuscript is good enough to be published.